# Design and Development of a Bimodal Optical Instrument for Simultaneous Vibrational Spectroscopy Measurements

**DOI:** 10.3390/ijms23126834

**Published:** 2022-06-20

**Authors:** Laura A. Arévalo, Stephen A. O’Brien, Eneko Lopez, Gajendra Pratap Singh, Andreas Seifert

**Affiliations:** 1Nanoengineering Group, CIC nanoGUNE BRTA, 20018 San Sebastián, Spain; l.arevalo@nanogune.eu (L.A.A.); obries39@tcd.ie (S.A.O.); e.lopez@nanogune.eu (E.L.); 2Disruptive & Sustainable Technologies for Agricultural Precision, Singapore-MIT Alliance for Research and Technology, Singapore 138602, Singapore; 3IKERBASQUE, Basque Foundation for Science, 48009 Bilbao, Spain

**Keywords:** Raman spectroscopy, FTIR spectroscopy, optical design, molecular fingerprint, biochemical analysis, Raman depth profile, optical sectioning

## Abstract

Vibrational spectroscopy techniques are widely used in analytical chemistry, physics and biology. The most prominent techniques are Raman and Fourier-transform infrared spectroscopy (FTIR). Combining both techniques delivers complementary information of the test sample. We present the design, construction, and calibration of a novel bimodal spectroscopy system featuring both Raman and infrared measurements simultaneously on the same sample without mutual interference. The optomechanical design provides a modular flexible system for solid and liquid samples and different configurations for Raman. As a novel feature, the Raman module can be operated off-axis for optical sectioning. The calibrated system demonstrates high sensitivity, precision, and resolution for simultaneous operation of both techniques and shows excellent calibration curves with coefficients of determination greater than 0.96. We demonstrate the ability to simultaneously measure Raman and infrared spectra of complex biological material using bovine serum albumin. The performance competes with commercial systems; moreover, it presents the additional advantage of simultaneously operating Raman and infrared techniques. To the best of our knowledge, it is the first demonstration of a combined Raman-infrared system that can analyze the same sample volume and obtain optically sectioned Raman signals. Additionally, quantitative comparison of confocality of backscattering micro-Raman and off-axis Raman was performed for the first time.

## 1. Introduction

For many years, vibrational spectroscopy has proven to be a powerful and fundamental technique for chemical analysis and has been applied to a broad variety of samples. Vibrational spectroscopy techniques detect the characteristics of the fundamental vibrations of molecules, or their “molecular fingerprint”, and thus the molecular composition of the analyte under analysis. That way, the chemical structure information of any compound is revealed in a qualitative and quantitative manner. The general principle of spectroscopy lies in the interaction of photons with chemical bonds of a sample that results in the generation of a spectral fingerprint with relevant information. Over many years, the establishment of sampling methods, operating procedures, data analysis, and validation has made such techniques a forerunner in determining the chemical structure of almost any substance.

The two main vibrational spectroscopy techniques are Raman and infrared (IR) absorption spectroscopies. Raman spectroscopy is based on inelastic light scattering, requires changes in the polarizability of molecules, and generally uses monochromatic radiation as a light source. Infrared absorption spectroscopy, mostly carried out in the form of Fourier-transform infrared (FTIR) spectroscopy, records the light absorption of a sample produced by the atomic and molecular motion caused by the interaction with infrared radiation [1] and requires changes in the dipole moment of the molecules. The application of both techniques, in combination or independently, has increased exponentially in numerous fields due to several specific advantages, as for example, spatial resolution in the micron range, specific biochemical information revealed without using labels, little or no sample preparation, no sample destruction, contactless operation, easy sample handling, real-time data acquisition and analysis, and low limit-of-detection. Raman spectroscopy can be carried out both with dispersion as well as interferometry (Fourier-transform) based instrumentation. Raman and FTIR are considered as complementary techniques, and both techniques have their individual strengths, as well as weaknesses, in relation to the nature of the sample, the sample handling, and the aim of the research. The study of any sample with both techniques results in expanded spectral information and more precise results, especially in cases where high sensitivity and specificity are required.

Analytical studies with vibrational spectroscopy include the characterization of polymeric structures and quantitative evaluation of several physical properties [2], identification of pharmaceutical drugs [3], quantification of contaminants in the environment [4], food control [5], forensic evidence analysis [6], authentication and dating of cultural heritage [7], detection of microplastics in ecosystems [8], medical diagnosis [9,10,11], and generally speaking chemical analysis.

In the field of medical diagnostics and biomedical topics, vibrational spectroscopy has proven its value in combination with multivariate analysis and chemometrics and is now stepwise entering into clinical applications [12]. The ability of detecting biochemical changes in cells and tissues allows for real-time disease diagnostics. Several excellent reviews demonstrate the utility of vibrational spectroscopy for in vivo, ex vivo, and clinical diagnoses. Balan et al. demonstrated the potential of such technology to be applied in clinical practice as a future standard [11]. Many studies reported in the literature have focused on the analysis of biological fluids, animal and human tissue, and cells. Biological fluids such as blood [13], saliva [14], cerebrospinal fluid [15], tears [16], synovial fluid [17], and semen [18] among others, are used to identify the health status of a patient or the presence of microorganisms, infections, or abnormal substances. Further studies on tissues and cells were carried out to unveil differences between malignant and benign cells to classify or diagnose different types of cancer [19,20,21,22,23,24].

Since Raman and FTIR spectroscopies deliver complementary information about biochemical composition and structure, the combination of the two modalities may offer higher sensitivity and specificity in disease prediction than simply using each modality alone. Many authors have reported on the use of both techniques for increasing the reliability of results and the robustness of the method as a whole. For example, Carmona measured blood samples of patients with Alzheimer’s disease as a method for diagnosing the disorder [25]. The use of two separate commercial instruments to solve the same problem implies many challenges. Splitting the sample is a common strategy employed in many studies utilizing the combination of Raman and FTIR techniques. This approach entails several problems and is likely to fail in scenarios where only small sample volumes are available, when identical measurement conditions are required, or in cases of time-dependent studies.

A rather large number of commercial Raman and FTIR spectroscopy systems with excellent performance, robustness, and user-friendly software are available on the market. In many cases, the commercial instruments are designed to solve specific experimental problems, thus restricting their applicability. Customized systems are more flexible since they can be modified to solve different types of problems by adapting them to specific requirements. Most of these instruments on the market provide only one technique and do not have the ability to measure with both techniques simultaneously. ThermoFisher Scientific, Waltham, MA, USA developed an instrument, called Gemini Analyzers, which can measure Raman and FTIR spectra. The instrument has two different sample containers, one for each technique, which does not allow for the use of both techniques simultaneously. It is primarily designed for the identification of explosives due to its portability; however, its performance in terms of resolution, spectral range, and sample type is limited. Recently, Photothermal Spectroscopy Corp., Santa Barbara, CA, USA has introduced a commercial instrument based on the optical photothermal infrared spectroscopy technique that can record IR and Raman spectra with submicron spatial resolution at the same spot of the sample. The instrument allows for vibrational spectroscopy studies of different types of samples, such as microelectronic devices [26], microparticles, or single biological cells. This instrument uses a bulky tunable pulsed laser that hinders field applications and is also cost-prohibitive, making it available only to large technological centers or institutions.

In this work, we present a combined instrument that carries out both methods, Raman and FTIR, simultaneously. The setup can measure a sample with both methods at the same time and under the same conditions, providing multiple advantages for any study. The design of this novel bimodal instrument is subsequently described in detail in Section 3. Raman and FTIR spectroscopies measure similar molecular vibrations, but the instrumentation design for their combination is nontrivial. The technical aspects and their components have been carefully chosen to optimize the quality of the measured spectra. Furthermore, the optomechanical design facilitates measurements with both techniques without mutual interference.

In the following section, we present the design of our novel bimodal vibrational spectroscopy system; then, we demonstrate the system’s performance with specific examples that address the utility of the combined instrument. In Section 3, we introduce in detail the technical specifications of Raman and FTIR. To the best of our knowledge, this article describes for the first time the design and development of a combined Raman–FTIR system where the same sample can be measured by both techniques and where optically sectioned Raman signal is obtained from the sample. Quantitative comparison of confocality of a traditional micro-Raman system and an off-axis excitation Raman system has also been performed for the first time.

## 2. Results and Discussion

### 2.1. Optical Design of the Combined System

Raman and FTIR spectroscopies are complementary techniques that measure vibrational modes of molecules but are based on fundamentally different physical phenomena. Each technique requires distinct instrumentation to acquire the spectra. The basic component in both spectroscopic techniques is a wavelength-dispersive device that selects narrow bands of wavelengths [27]. In the case of the FTIR technique, this process is performed with an interferometer, whereas for Raman, the mechanism used is a diffraction grating-based monochromator. In FTIR spectroscopy, light from an IR source enters the interferometer, produces an interference pattern, and then interacts with the sample before being projected onto an IR detector. In Raman spectroscopy, a laser beam is focused onto or into the sample, and the inelastically scattered Raman signal is collected and imaged onto the monochromator, which directs the dispersed radiation onto a detector [1].

Figure 1 presents a schematic of our experimental setup featuring both Raman and FTIR spectroscopy techniques. Two different configurations of the Raman system are shown, an upright configuration (Figure 1a) and an off-axis inverted configuration (Figure 1b). Both geometries can be easily interchanged. The orange lines represent the beam path of the FTIR, while the blue and red lines represent the Raman beam path. The FTIR system is designed for ATR (attenuated total reflectance) modality as a sampling method. The common optical element for both techniques is the ATR crystal on which the sample is placed. The Raman system is designed to operate on a micro and macro scale, depending on the needs of the user.

The infrared source and the interferometer are situated in the left compartment of the FTIR setup. Light is guided by paraboloidal mirrors to the ATR crystal, which is used as a waveguide in total internal reflection mode. In the ATR mode, a small fraction of the light interacts with the sample at the upper interface of the crystal through evanescent waves for each reflection. The light that is not absorbed by the sample is detected by an infrared detector. Again, paraboloidal mirrors are used [28,29] to guide the light onto the detector, as shown in our schematic on the right-hand side.

The Raman design consists of an excitation arm (red lines) and a collection arm (blue lines). The excitation arm consists of a laser, a band pass filter, and optical elements for expanding and focusing the light into the sample. The collection arm uses optical elements to collect the Raman signal and to send it into the spectrometer, as well as a long pass filter. In a typical Raman setup, both arms share the beam path until a dichroic filter separates them; this configuration is called the 180° scattering geometry [30]. Although this configuration is robust in its alignment, interferences from scattering and fluorescence generated in the optical components can occur.

An alternative geometry is also proposed in this setup, an off-axis or oblique illumination where the excitation arm forms an angle α with the optical axis of the observation arm (Figure 1b); this geometry reduces the signal of the ATR crystal in the recorded spectrum and permits the use of different lenses for the illumination and observation, allowing for optical sectioning under certain circumstances [31]. Furthermore, no dichroic mirror is needed in the oblique configuration, optimizing the light collection by eliminating an additional optical element.

The path of some of the beams within the ATR crystal is shown in Figure 1c,d for both FTIR and Raman techniques. In this case, a trapezoidal crystal of 9 × 1.5 mm is simulated in the ray tracing analysis, which produces three internal reflections of an infrared beam (λ = 5 µm) that enter through the lateral side of the crystal at 45°. The maximum cone of light for the Raman signal collected with the inverted configuration is also shown. The configurations of each of these techniques is described in more detail in Section 3.

### 2.2. Raman Depth Profiles

The concept of optical sectioning for our Raman system is demonstrated experimentally in this section by using our custom bimodal system. A Raman intensity depth profile is generated by focusing the laser beam onto a surface of a transparent or semitransparent sample and moving the object under study away from the focal plane in the direction of the optical axis, thereby moving the focus into the sample incrementally as shown in Figure 2a. A Raman signal is acquired for each position *z*, and thus, plotting the Raman intensity of one specific band versus the distance *z* from the sample surface reveals structural depth information of the sample.

The ability to map 3D materials with a high spatial resolution is one of the most valuable features of a Raman system. The lateral spatial resolution of a Raman system can be determined by the Airy disc [32]; however, the depth resolution cannot only be described with the Airy disc; it depends on the detection optics as well as the illumination optics, as many authors have shown [33,34,35]. A depth resolution for a microscope Raman system has traditionally been evaluated using a thin film as an evaluation object in a Raman depth profile measurement [36]. The full width at half maximum (FWHM) of the confocal Raman response provides insight into the confocal behavior of the system. The Raman volume, i.e., the volume from which the Raman signal is collected by the system, depends on the numerical aperture of the detection and illumination optics, as well as on the wavelength, confocal aperture, and refractive index of the sample. This last parameter will strongly determine the depth profiles, as Everall demonstrated by showing how depth resolution degrades when focusing deeper within a sample [33].

The Raman depth profile was investigated with a silicon wafer using both configurations, confocal micro-Raman (Figure 2b), and oblique illumination (Figure 2c), however without the ATR crystal. The depth profiles were obtained following the procedure described in [36], i.e., a silicon (Si) wafer with a thickness of 550 µm was used as a stable reproducible object for inspection. The wafer was moved step by step about the focal plane and along the optical axis of the collection lens, as Figure 2a shows. At each step, a Raman signal was taken, and as a result, the depth profile was assembled using the counts of the maximum peak of the Si Raman signal near 519 cm^−1^.

For the micro-Raman configuration, a microscope objective plan-neofluar of 63X/0.75NA (Zeiss-421380-9971-000) was used. The confocal aperture was set with multi-mode fibers of 105, 200, and 400 µm and with a bundle of eight fibers of 400 µm core each. Each single spectrum was recorded with an acquisition time of 1 s for each 2 µm shift in *z*. The position of the Si wafer where the maximum Raman signal is achieved is defined arbitrarily as *z* = 0. It is important to note that *z* = 0 is not necessarily the position of the focus on the sample surface [36]. The resulting depth profiles for the silicon wafer are shown in Figure 3a. The curves were normalized in order to visualize the differences in the profile for each fiber used; however, when a smaller fiber is used, the intensity of the Raman signal decreases; thus, a trade-off between the signal intensity and the axial resolution should be taken into account in each experiment.

The FWHM of these curves are an indicator of the depth resolution of the system for each specific sample [37]. In the case of Figure 3a, the FWHM is found to be 44 µm for the smallest fiber core size of 105 µm, and 75 µm for the bundle of eight fibers of 400 µm core each. The results show that the smallest fiber captures signals from a small volume, as expected from a confocal configuration. It can be concluded that the efficiency (signal-to-noise ratio SNR) of collecting a small volume by the collection optics is better with a small fiber since it rejects the contribution of the volume surrounding the Raman signal collection volume.

The inverted Raman system was tested for both configurations, 180° backscattering geometry and oblique illumination. The depth profiles were measured with an observation (collection) lens of numerical aperture NA = 0.13, and in the case of the oblique configuration, the illumination lens has an NA of 0.17 due to geometrical restrictions. The angle in the oblique setup was 45° and the *z*-shift for the depth profiling was 10 µm. Figure 3b shows the resulting profiles for the two geometrical configurations, where a clear enhancement of the depth resolution is obtained when the oblique illumination is used. For the 180° geometry, the FWHM is 2400 µm, and for oblique configuration, the FWHM is 400 µm. The better depth resolution in off-axis geometry is a result of a smaller sampling volume, as illustrated in Figure 2c. The configuration with a large Raman volume can be useful in the case of homogeneous samples such as liquids, where large integration volumes are considered.

A further comparison between the two geometries, oblique and 180°, was carried out by measuring a thin film of PMMA (polymethyl methacrylate). The film of 50 µm thickness was placed on top of the ATR diamond crystal. Figure 4 shows the spectrum of the PMMA taken with the two collection/illumination geometries. One sees that the Raman spectrum of the PMMA is strongly affected by the Raman signal of the diamond, which has a strong main peak at 1333.1 cm^−1^ and dominates the Raman signal of the film. A rescaling allows the bands of the PMMA film to be seen. The diamond peak appears stronger in the case of the 180° configuration because a larger volume of the diamond is excited and collected (Figure 2b). Furthermore, the depth profile for the 180° configuration shows that the Raman volume is in the order of millimeters (Figure 3b), causing part of the sample signal and part of the crystal signal to be captured. However, in the oblique configuration, where a quite large volume of the diamond is illuminated and the corresponding Raman signal is generated, this Raman signal is not collected by the collection lens, as shown in Figure 2c. As can be seen in Figure 3b, the Raman volume for the oblique configuration is smaller than for the 180° configuration, resulting in strongly reduced interferences from the diamond, and from the sample itself, thereby changing the dynamics of the signal because of a reduced background.

This advantage of optical sectioning in off-axis geometry can be seen in the spectrum of Figure 4, where some of the bands are better defined in the oblique configuration than in the 180° geometry, as for example the Raman band around 370 cm^−1^ and the PMMA Raman band overlapping with the diamond Raman band.

### 2.3. Vibrational Spectroscopy of Chemical Compounds

We demonstrate the performance of the custom-built FTIR–Raman system by the spectra of reference chemical compounds. The viability of recording vibrational spectra from solutions was evaluated using sodium lactate diluted in Milli-Q water at different concentrations. The measurement of the sample at different concentrations allows us to produce a calibration plot that reflects the relationship between the intensity of the spectra and the concentration, demonstrating the potential of the technique as an analytical method for quantitative analysis. To test the sensitivity and calibrate the combined FTIR–Raman system, a test of the Raman and FTIR spectra was carried out using molar concentrations of sodium lactate ranging from 0 to 5.0 M in steps of 0.5 M. For the calibration measurements, the inverted geometry was used with a 45° off-axis angle and an observation lens with effective focal length of 12 mm and NA of 0.38 as well as an illumination lens of effective focal length of 18 mm and NA of 0.25. For each concentration, 2500 µL of the analyte was placed onto the ZnSe ATR crystal (which has multiple reflections and hence greater sensitivity), and the chamber was covered with a glass slide to avoid sample contamination. The simultaneous measurements were then carried out for each technique as described below.

#### 2.3.1. FTIR

As shown in Figure 5, characteristic peaks in the fingerprint region of an analyte show systematic changes with the concentration, which can be positive or negative. The peak variations allow us to generate a calibration plot where the intensity changes and the concentration are linearly related following Lambert-Beer’s law
(1)A=εlc,
where *A* is the absorbance, ε the absorptivity, *l* the path-length, and *c* the concentration.

For the FTIR measurements, a spectrum with the clean ATR crystal was performed to calculate the absorbance. An average of 10 spectra were measured using medium (Norton-Beer) interferogram apodization in a sequence measurement with five total measurements and a sequence pause of 1 ms. After each measurement, the crystal was cleaned with distilled water followed by 100% isopropanol. Two repetitions were carried out for each of the concentrations measured, giving a total of 10 measurements per concentration.

Figure 5a shows the FTIR spectra for sodium lactate concentrations in the range of 0.0–5 M. The figure also shows the chemical structure and a stick-and-ball figure of lactate. Figure 5b presents the calibration curves of the peak intensities at 1040 cm^−1^ (black), 1120 cm^−1^ (red), 1314 cm^−1^ (blue), and 1416 cm^−1^ (green) as a function of the sodium lactate concentration. All peaks display excellent linear trends with increasing molar concentration and coefficients of determination R^2^ around 0.999, which is consistent with Equation (2). The calibration parameters of the linear regression are given in Table 1, along with the limit of detection (LoD) for each vibrational peak, demonstrating the ability to measure the analyte with concentrations as low as 0.0297 M.

#### 2.3.2. Raman

The general expression for the Raman intensity as a function of the number of vibrating molecules *N* is given by
(2)$$I{\left(N\right)}_{R}\propto NI_{0}\nu^{4}{\left(\frac{\partial \alpha }{\partial Q}\right)}^{2},$$
where *I*_0_ is the intensity of the incident laser beam, *ν* is the excitation frequency, and (∂*α*/∂*Q*) is the Raman cross-section in terms of the molecular polarizability α and the vibrational amplitude *Q* [1]. The Raman cross-section is a constant of the analyte, and the excitation frequency remained constant for all measurements; thus, the intensity of the Raman scattering signal is linearly proportional to the concentration of the analyte.

Calibration measurements for the Raman spectra were carried out simultaneously with FTIR using the same sodium lactate solutions. Two samples of each concentration were used, and an average of 10 Raman spectra was measured with an integration time of 1 s. The Raman intensity for each concentration was averaged, and the peak intensities of selected peaks of sodium lactate were plotted as a function of the molar concentration.

Figure 6a presents the Raman spectra for various concentrations of sodium lactate in the range of 500–1200 cm^−1^, which display the main vibrational modes. Figure 6b shows the calibration curves of the Raman peak intensities at 856 cm^−1^ (black), 1043 cm^−1^ (red), and 1086 cm^−1^ (blue). Consistent with Equation (2), these peak intensities follow a linear trend with increasing molar concentration. The parameters of the linear regressions are included in Table 2, and all lines exhibit coefficients of determination greater than 0.96, which demonstrate excellent agreement of the data with a linear trend. The LoD for each vibrational peak is also presented, demonstrating the ability to detect analytes with concentrations as low as 0.05 M.

Both calibration plots demonstrate that the combined FTIR–Raman system can precisely measure changes in molar concentration with excellent sensitivity on par with commercial systems. As an additional asset, our system is able to simultaneously measure FTIR and Raman spectra, and it can carry out Raman measurements in a novel off-axis geometry, performing optical sectioning through the crystal.

### 2.4. Vibrational Spectroscopy of Complex Compounds

Mixtures of multiple components or complex molecules with a broad range of functional groups can produce complex spectra from the combination of each characteristic band. For example, highly complex spectra can come into view when biological samples are analyzed because many different types of biomolecules and chemical bonds are involved, such as proteins, lipids, carbohydrates, nucleic acids, among others. Particularly, protein analysis has become an essential tool in the biomedical field, as it can indicate the presence of specific pathologies or the physiological state of a patient.

One of the most important structures of proteins are their secondary structure, which include polypeptides such as α-helix and β-sheets. These structures are mainly formed by hydrogen bonds between carbonyl oxygen atoms and amino hydrogen atoms in amide bonds. These amide vibrational bands can be described in terms of five in-plane (C=O stretching, C–N stretching, N–H stretching, OCN bending and CNH bending) and three out-of-plane (C–N torsion, C=O and N–H bending) displacement coordinates [38].

Changes in the conformation of proteins can affect physiological functions and cause disorders, as in the case of degenerative diseases of the central nervous system, such as Alzheimer’s disease [39]. Amyloid-beta 42/40 (Aβ40/42) and tau neurofibrillary tangles have been generally accepted as biomarkers of Alzheimer’s disease [40]. These biomarkers can be found in some biofluids such as cerebrospinal fluid (CSF) or blood, and the structural conformation of these can be studied with vibrational spectroscopy, as many authors have demonstrated in the past [25,41,42].

In this section, the capability of the bimodal instrument to analyze complex samples, such as CSF, is demonstrated. For this purpose, we studied with our instrument a protein mixture from bovine serum albumin (BSA). BSA is a globular protein widely used in protein studies; it was purchased from Sigma Aldrich and prepared with phosphate-buffered saline (PBS). Typical protein concentrations in CSF of healthy adults fluctuate about an average of 0.50 mg/mL [43]. In vibrational spectroscopy studies with CSF, the sample is filtered by an ultra-centrifugation process developed by Bonnier [44], this procedure increases the concentration of the proteins in the sample. An initial volume of CSF of 500 µL is centrifuged, resulting in a final volume of 15 µL with high protein concentration of 16 mg/mL. In our BSA model, we used this exemplary volume and concentration and deposited it onto the diamond ATR crystal and spread it evenly using a spatula. Due to the small volume of the sample, the small diamond ATR crystal was used. The drop was allowed to air-dry for 30 min to avoid interference of water spectral bands with BSA spectral bands.

The FTIR spectra of dried BSA were acquired with 4 cm^−1^ resolution and an average of 32 scans. Raman spectra were taken with 4 cm^−1^ spectral resolution with a laser power of 20 mW, 100 acquisitions, and with an exposition time of 1 s. A 63x (NA 0.75) microscope objective was used to collect the signal due to the weakness of the Raman signal generated in this sample. The IR absorbance spectrum and the Raman spectrum of a BSA sample in the region of 3700–800 cm^−1^ are shown in Figure 7. For noise reduction, both spectra were smoothed with a Savitzky–Golay filter of 7-point window and third-degree polynomial sampling at 3.67 cm^−1^. Additionally, the Raman spectra were baseline corrected by means of asymmetric least squares (ALS) fitting. In the Raman spectrum, the diamond band can be seen interfering with the amide III band; however, the other bands are seen clearly. In the FTIR spectrum, the region between 1800 and 2600 cm^−1^ (not shown) is dominated by noise due to the low optical transmission of the diamond in that region.

According to the literature in [45], the IR absorption bands of BSA, labeled in Figure 7, are mostly assigned to the vibration of the amides. The amide A band centered around 3289 cm^−1^ is fairly broad due to the stretching of the nitrogen and hydrogen bond of the amide. The amide B band results from a similar stretching mode (N–H stretch). The amide I band at around 1650 cm^−1^ results from two vibrational modes, a stretching of the carbon oxygen double bond that accounts for 70–80% of the absorption, and a stretching of the carbon and nitrogen bond that accounts for the remaining 10–20% of the absorption (C=0 stretch, C–N stretch). The amide II band centered at 1542 cm^−1^ results from three modes, a bending of a nitrogen and hydrogen bond (40–60%), a stretching of a carbon–nitrogen bond (18–40%), and a stretching of a carbon–carbon bond (about 10%), N–H bend, C–N stretch, and C–C stretch. The final band shown is the amide III band that results from a complex mix of several vibrations.

The Raman spectrum contains several peaks associated with proteins and amino acids that include tyrosine (Tyr), phenylalanine (Phe), tryptophan (Trp), amide III and am-ide I. These bands have previously been reported in the literature [25,41]. The vibration of the amide I appears with a maximum at 1658 cm^−1^, this maximum intensity can be assigned to a α-helical protein structure. The band in the region 760–740 cm^−1^ can be attributed to the indole-ring breathing mode of tryptophan side chain (Try). The relevance of these bands was demonstrated for medical diagnostics of Alzheimer’s disease, where a difference in the Raman spectrum of patients with the disease in relation to a healthy control group by means of machine learning algorithms was reported [42].

With this example, we have demonstrated the high capacity of the instrument to be applied in complex problems such as medical diagnostics; in particular, we showed the complementarity in biochemical information delivered by these two methods. Due to its portability, our bimodal instrument has the potential to be applied in real-life scenarios, such as clinical settings. Our bimodal vibrational spectroscopy instrument allows for a complete in-depth study of biochemical composition of any sample. In the future, machine learning and artificial intelligence-based algorithms can be used to combine information from FTIR and Raman spectra and to develop robust models for disease diagnostics.

## 3. Materials and Methods

### 3.1. FTIR Module

An FTIR spectrometer is based on an interferometer, such as the Michelson interferometer shown in Figure 1a. A beam-splitter (BS) is used for dividing the light into two beams. Each beam is reflected onto a mirror: one of the mirrors is fixed (FM), while the other is movable (MM). The two beams are recombined by the same beam-splitter and are sent to the sample chamber. If the optical distance between the two mirrors and the beam splitter is the same, both beams travel the same distance before reaching the sample and the detector. It is said that the two beams are in phase. However, when the movable mirror moves, one of the beams will travel an extra distance, and a retardation or optical path difference between the two beams will be produced. The recombined light leaves the interferometer and is directed to the ATR crystal, where the sample absorbs part of the light from the evanescent field. The light that is not absorbed is redirected toward the IR detector through some optical components such as paraboloidal mirrors (PM). For each step of the movable mirror, the detector measures the interference of light, obtaining a curve of intensity versus retardation, which can then be translated into spectrum versus wavenumber through a mathematical operation called Fast Fourier Transformation. An infrared spectrum can be represented by an absorbance spectrum, which measures the amount of light absorbed by a sample and is calculated from the background spectrum (spectrum without the sample) and the sample spectrum following
(3)$$A=\log\left(\frac{I_{0}}{I}\right)=\log\left(\frac{1}{T}\right)$$
where *A* is the absorbance, *I*_0_ the intensity of the incident light, *I* the intensity of the light after it has passed through the sample, and *T* the transmittance, which measures the amount of light transmitted through a sample [28].

The hardware used in our FTIR setup was manufactured by ARCoptix S.A, Neuchatel, Switzerland and consists of the following modules: interferometer, detector and light source [46]. The interferometer module, called FTIR “Rocket” OEM module, uses a so-called Globar as IR source, which is a thermal light source for infrared spectroscopy based on silicon carbide rods. A Globar lamp covers a wider spectral range than halogen lamps; the spectral range of the lamp in wavenumbers is 9000–200 cm^−1^. The interferometer is based on a dual corner-cube (retroreflector) design, where two plane mirrors are mounted at 90° over a common arm that rotates to create the retardation between the two beams. With this design, the effect of tilting errors in the Michelson interferometer is minimized, and the alignment is mechanically more robust and durable. The beamsplitter material is ZnSe, and the instrument operates with a spectral resolution of 4–8 cm^−1^; acquisition velocity is 1 spectrum per second; output infrared beam has a divergence of ~28 mrad and an aperture of ½ inch. The detector, a thermoelectrically cooled MCT (HgCdTe), can measure over a spectral range of 5000–700 cm^−1^, corresponding to a wavelength range of 2–14 µm. This range is ideal to identify biomarkers for different biological samples. The fingerprint region, located within the MIR (mid-infrared) region between 8.3–14 μm (1200–700 cm^−1^), contains bands from lipids, proteins, carotenoids, and polysaccharides, and as a result is rich in structural information.

As a sampling technique, a reflection mode is employed using an ATR crystal. This contact sampling method involves a crystal with a high refractive index and excellent transmittance of IR radiation. We use two types of crystals, a diamond crystal with a parallelogram shape angled at 45° and a ZnSe crystal with the same shape but greater length and hence a greater number of internal reflections. Diamond is characterized by its exceptional hardness, robustness, durability, and its optical and thermal properties. It is inert, tolerating pH ranges from 1–14, and it has a refractive index ranging from 2.406 at 700 nm, to 2.375 at 14 μm. One great advantage of this material is the sample handling; residues on the surface can be easily removed using chemical compounds, and even sterilization at high temperatures can be performed. The penetration depth of the evanescent wave is approximately 2 µm, depending on wavelength, and increases with decreasing wavenumber. The dimensions of the crystal, 9.0 × 4 × 1.5 mm^3^, allow for three internal reflections or contact points with the sample (Figure 1c). The diamond crystal is attached to a metallic holder using active brazes and was made by Dutch Diamond Technologies, Cuijk, The Netherlands [47]. One of the disadvantages of diamond is its high cost. Alternatively, we also use a ZnSe crystal, which is one of the most used materials in ATR technology due to its low cost and low toxicity compared to other materials. In contrast to diamond, ZnSe is highly fragile and susceptible to scratching. The ZnSe crystal has a trapezoidal prism geometry of size 52 × 20 × 2 mm^3^ with an angle of 45° at the entrance facet; it produces about 13 internal reflections, which makes it perfect for high-sensitivity measurements. Exchanging the crystals requires an adjustment process each time.

The collimated light that emerges from the interferometer is focused onto the entrance facet of the ATR crystal and is refocused onto the detector after exiting the crystal. In our customized optical design, the beam is guided by off-axis paraboloidal mirrors to avoid spherical and chromatic aberrations, while at the same time minimizing losses. Two off-axis paraboloidal mirrors are used, with focal lengths of 203.2 mm, off-axis angles of 45°, and with unprotected gold coatings, as shown in Figure 7. To avoid losses, we chose the unprotected gold mirrors with maximum losses of around 8%.

### 3.2. Raman Module

Due to the underlying physics and low scattering cross section, the Raman technique operates with weak signal intensities, which makes an appropriate choice of instrumentation and careful optical design vital. Figure 1 shows a schematic of the Raman system consisting of an excitation arm (red lines) and a collection arm (blue lines). The excitation energy comes from a single frequency diode pumped laser (DPL) of 785 nm, this wavelength minimizes the fluorescence and the thermal decomposition of the sample due to lower photon energies. However, the Raman efficiency drops strongly in comparison with visible excitation that is inversely proportional to the 4th power of the wavelength of the scattered light. The laser light is guided toward the setup through a single mode optical fiber via an FC/PC connector. The fiber is coupled to a collimator, which is a system of a triplet lens that provides nearly a Gaussian output, optimized for 780 nm, with a focal length of f = 6.06 mm and a numerical aperture of NA = 0.28. A bandpass filter spectrally cleans the laser beam in the collimated part by selecting a narrow band of wavelengths from the emitted spectrum. As optical material, fused silica is used, as it is transparent over a wide range of wavelengths (185 nm–2.1 µm) and produces minimum fluorescence background.

The excitation and collection lenses focus the light onto the sample and collect the Raman signal, respectively. Another function of the collecting lens is to collimate the light for efficient filtering afterward. In the case of a 180° scattering geometry (Figure 1a), the excitation and collection lenses are the same; a microscope objective can also be used to conduct micro-Raman measurements. In the case of the oblique configuration (Figure 1b), these lenses can be easily exchanged to have different numerical apertures, depending on the specific application.

A subsequent filter in the optical path of the collection arm is designed to transmit wavelengths greater than the cut-off wavelength of the filter, which in this case is 785 nm (Ref. LP02-785RU-25 supplied by Semrock, Rochester, NY, USA) to suppress unnecessary signals of high intensity originating from the light source, scattering from the sample, or indirect unwanted reflections within the optical system. An achromatic lens is used to focus the parallel beam onto the collection fiber, which is connected to the spectrometer. Achromatic doublets are useful for controlling both chromatic and spherical aberrations and are generally used to achieve a diffraction-limited spot when using a monochromatic source such as a laser. Our lens (Ref. AC127-019-AB-ML supplied by Thorlabs, Newton, NJ, USA) has an anti-reflection coating for the spectral range of 400–1100 nm and a focal length of f = 19.0 mm. The spectrometer used is an EAGLE Raman-S manufactured by Ibsen Photonics, Farum, Denmark, and has a resolution of 4 cm^−1^, spectral range of 3685–133 cm^−1^, F-number of 1.6, 20 µm slit width, diffraction grating with 1500 lines/mm, TE-cooling down to –60 °C and a pixel number of 2000 × 256 (15 × 15 μm).

Some samples, especially biological ones, suffer from weak Raman signals that are hidden due to a large background signal. Although the removal of background can be solved a posteriori using analytical data processing techniques, an experimental and practical solution is necessary to guarantee a good quality of the Raman spectra [48]. To isolate the sample signal from other signals coming from elements that could interfere, such as the substrate, which in our case is the ATR crystal, many strategies can be employed to overcome this problem, such as a confocal configuration or an off-axis illumination arm, as we propose in this work.

A confocal Raman configuration requires a pinhole, which cuts spatial parts of the signal and allows for the recording of only the light that comes from a specific point or depth range. In the case of Raman microscopy, the input of the fiber that transmits the light to the spectrometer is used as the pinhole; thus, no additional opto-mechanical components are required. The size of the fiber determines the degree of confocality: the smaller the fiber, the better the confocality, and hence optical sectioning, but the amount of light that enters the spectrometer is reduced. In practice, an appropriate fiber core size must be chosen to achieve an optimized signal-to-noise ratio with respect to the experiment [32].

A Raman off-axis system would allow for optical sectioning under certain circumstances. Combinations between the numerical aperture (NA) values of the excitation lens and the collection lens of the Raman system give flexibility to the setup. In particular, high values of NA for the excitation lens lead to excitation of small volumes, while lower values of NA lead to excitation of larger volumes. In turn, high values of NA for the collection lens lead to integration of information from small volumes, while lower values of NA lead to integration of information from extended volumes. An off-axis configuration has already been proven as a method for obtaining effective optically sectioned Raman images [31]. Some of the advantages of this geometry include spectra with high axial resolution, a decrease in the photodamage in the sample, and reduction of interferences due to the substrate. Depending on the state of matter of the sample under test, liquid or solid, and depending on the homogeneity of the sample, a choice of low or high numerical aperture will individually optimize the measurement result. Moreover, an oblique illumination can also be beneficial in the case of SERS where it has been demonstrated that the Raman signal is anisotropic, and an oblique illumination enhances the signal [49].

### 3.3. Sample Cavity and Mechanical Parts

To assure high flexibility, mechanical robustness and quick modifications of the functionality, a cage system is used for mechanical alignment and stability. The Raman module is small and rigid, which allows for flipping, such that the system can be operated top-down, as an inverted microscope or in the inverted off-axis geometry for optical sectioning.

Water vapor and carbon dioxide significantly interfere with the spectra of FTIR measurements. To overcome these problems, many instruments are purged with gases such as dry nitrogen. In our case, to ensure portability, the system is sealed and desiccated. A desiccant pack placed inside the housing of the instrument reduces water vapor and CO_2_ from the atmosphere surrounding the components. With dimensions and handling requirements in mind, a box enclosure to house the system was designed and fabricated. A photo of the completed instrument is shown in Figure 8a. The box is made of aluminum and has a door to access the instrument. Aluminum does not produce Raman signals, and by enclosing the system in this box, we avoid further external straylight.

The ATR crystal was manufactured by a company that grows synthetic diamonds of any dimension and geometry [47]. The crystal was fabricated with a metallic mounting as a stable and safe holder, as shown in Figure 8b. For biosafety reasons, the holder was designed with holes that allow the mechanical coupling of a cover plate that protects the sample and the user while measurements are taken; thus, any transfer of biological or hazardous material to the outside is avoided. The crystal holder acts as a little container for retaining liquid samples directly on the crystal.

## 4. Conclusions

We showed in detail the design and construction of a bimodal vibrational spectroscopy instrument featuring Raman and ATR–FTIR spectroscopies. The optomechanical design allows us to simultaneously operate both modalities without any mutual interference. The system is built up in a modular way, such that it can be flexibly adapted to several configurations: according to the needs, the Raman part can be used top down or as inverted microscope, and moreover, it can be operated in an off-axis configuration that allows for optical sectioning and flexible combinations of excitation and collection volumes.

With both techniques, we can quantify biochemical substances by highly linear calibration curves, with a performance comparable to commercial systems. By measuring complex biological samples, we demonstrated the complementarity of Raman and infrared absorption spectroscopies, which increases the information value in biochemical analysis or chemometrics and makes our combined system a useful tool for several applications in analytical chemistry, including studies of polymeric material such as microplastics for environmental monitoring, biological material for food quality control, or biofluids for in vitro medical diagnostics.

From an economic point of view, our proposed combined spectroscopy system stands out compared to commercially available instruments. The total costs of our combined instrument can be estimated to 60–70% of the costs for two individual instruments, which are an optical fiber-based portable confocal Raman spectroscopy system and a portable ATR–FTIR spectroscopy system. Moreover, an off-axis Raman system as proposed is not available off the shelf from commercial suppliers.

## 5. Patents

International application no. PCT/EP2021/062511 for “Combined spectroscopy system including Raman and ATR-FTIR”, in the name of Asociación Centro de Investigación Cooperativa en Nanociencias “CIC nanoGUNE”.

## Figures and Tables

**Figure 1 ijms-23-06834-f001:**
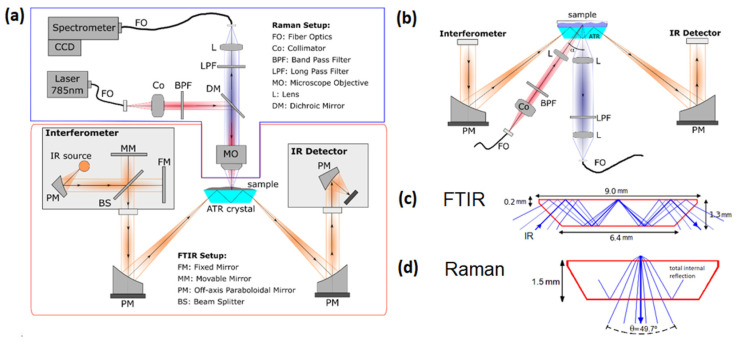
Schematic of the combined FTIR–Raman setup. (**a**) Upright configuration for micro-Raman. (**b**) Inverted off-axis configuration for Raman. Subfigures (**c**,**d**) illustrate ray tracing within the ATR crystal for both techniques: (**c**) Raman collection arm and (**d**) FTIR with three internal reflections.

**Figure 2 ijms-23-06834-f002:**
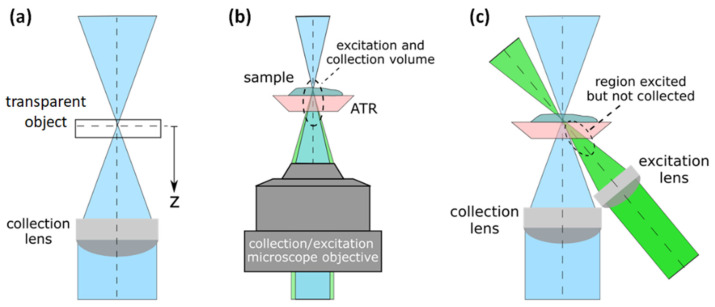
(**a**) Diagram of the Raman depth profile assembly. (**b**) Collection and excitation scheme with a microscope objective in the 180° geometry. (**c**) Collection and excitation scheme with different lenses in oblique configuration.

**Figure 3 ijms-23-06834-f003:**
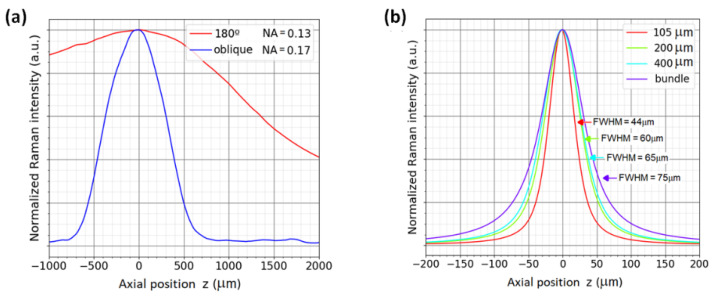
Comparison of Raman signal depth profiles for 519 cm^−1^ band of silicon wafer, thickness of 550 µm, measured with (**a**) micro-Raman confocal back scattering configuration with fibers of different core sizes acting as pinholes and (**b**) off-axis configuration capable of optical sectioning with 45° illumination compared to a traditional 180° configuration. Both systems utilize the same collection optics; the excitation optics differ slightly (180°: NA = 0.13, off-axis: NA = 0.17).

**Figure 4 ijms-23-06834-f004:**
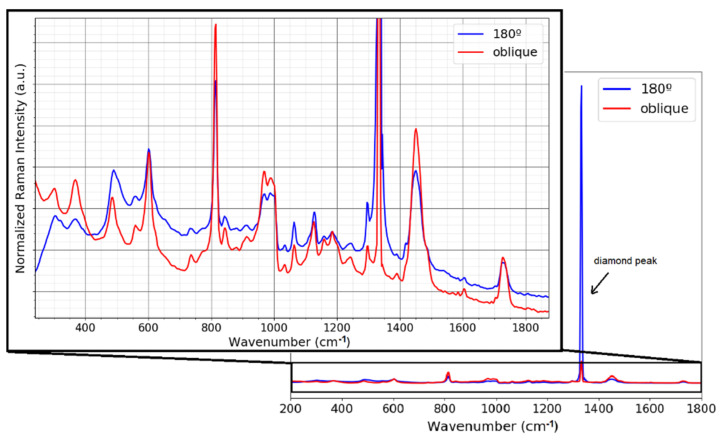
Raman signal of 50 µm PMMA thin film on top of the diamond ATR crystal. The right part shows the full spectrum with complete dynamics of the diamond peak, whereas the left figure zooms into the ordinate, making the much weaker PMMA Raman signals visible.

**Figure 5 ijms-23-06834-f005:**
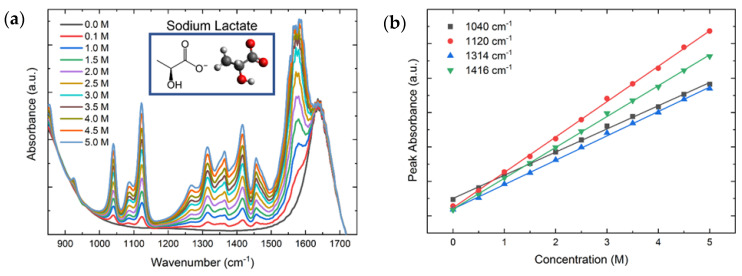
(**a**) FTIR spectra of sodium lactate for concentrations in the range of 0.0–5.0 M. (**b**) Peak intensities at 1040 cm^−1^ (black), 1120 cm^−1^ (red), 1314 cm^−1^ (blue), and 1416 cm^−1^ (green) as a function of the molar concentration. The peak intensities follow a linear trend with coefficients of determination around 0.999 for all peaks.

**Figure 6 ijms-23-06834-f006:**
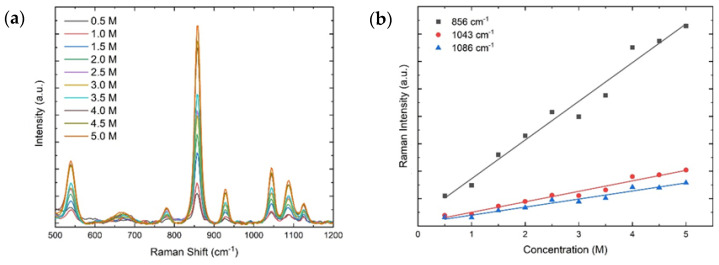
(**a**) Raman spectra of sodium lactate for concentrations in the range 0.0–5.0 M. (**b**) Peak intensities for 856 cm^−1^ (black), 1043 cm^−1^ (red), and 1086 cm^−1^ (blue) as a function of the molar concentration. The peak intensities follow a linear trend with coefficients of determination larger than 0.96 for all peaks.

**Figure 7 ijms-23-06834-f007:**
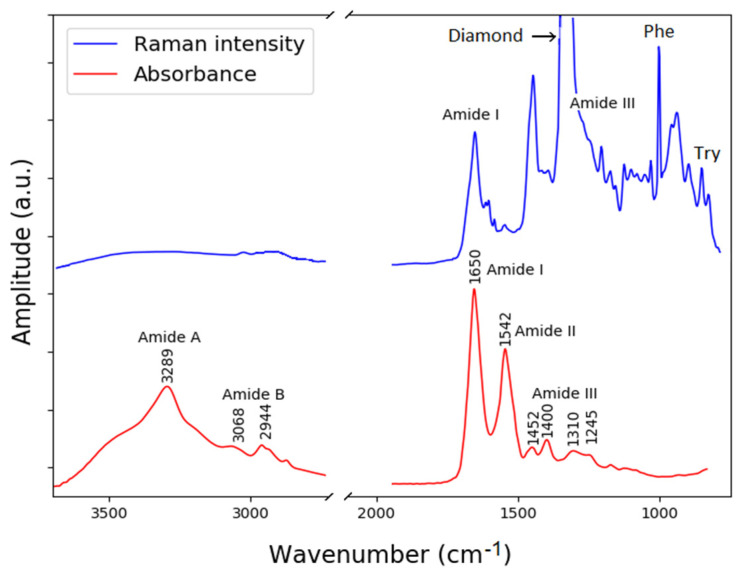
Raman and FTIR absorbance spectra from BSA acquired using the novel combined Raman−FTIR system. Corresponding vibrational bands are indicated.

**Figure 8 ijms-23-06834-f008:**
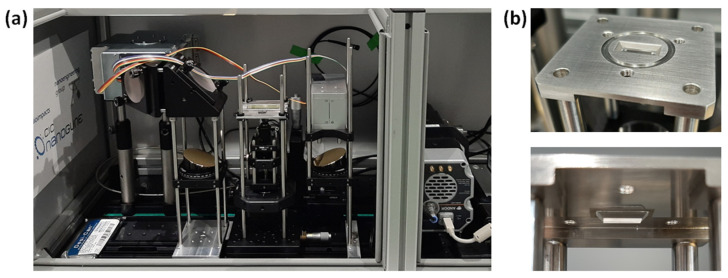
(**a**) Photograph of the combined FTIR–Raman setup with the Raman system in an inverted configuration. (**b**) ATR crystal with sample cavity from top and view from bottom.

**Table 1 ijms-23-06834-t001:** Linear regression coefficients of four FTIR peak intensities as a function of increasing molar concentration.

Peak (cm^−1^)	y-Intercept (a.u.)	Slope (a.u.) (cm)	R^2^	LoD (M)
1040	1.253 ± 0.011	0.3367 ± 0.0037	0.9988	0.0454
1120	1.116 ± 0.014	0.5142 ± 0.0046	0.9992	0.0297
1314	1.100 ± 0.011	0.3541 ± 0.0036	0.9990	0.0431
1416	1.253 ± 0.011	0.4485 ± 0.0039	0.9993	0.0341

**Table 2 ijms-23-06834-t002:** Linear regression coefficients of three Raman peak intensities as a function of increasing molar concentration.

Peak (cm^−1^)	y-Intercept (Counts)	Slope (cm)	R^2^	LoD (M)
856	310 ± 250	1411 ± 81	0.97	0.05
1043	117 ± 70	383 ± 23	0.97	0.18
1086	103 ± 63	293 ± 20	0.96	0.24

## Data Availability

Not applicable.

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
