# Peer review of "Design and Development of a Bimodal Optical Instrument for Simultaneous Vibrational Spectroscopy Measurements"

_ijms, 2022, doi:10.3390/ijms23126834_

Round 1
Reviewer 1 Report
I recommend the manuscript for publication. It deals with the combined maintenance for simultaneous registering Raman and infra-red spectra for chemical substances. The measuring system is well described and its work is examplified.
Author Response
Thank you very much for your review.
Reviewer 2 Report
The comments are on the attached file.

Author Response
Thank you very much for your review.

Reviewer 3 Report
In this work, the authors present very interesting results related to a combined instrument that carries out Raman and FTIR techniques at the same time and under the same conditions. Their way to explain all concepts is outstanding and the figures are wonderful. Congratulations.
Authors should go all over the text to correct some minor mistakes in written English. Also, they should use error bars in figures (5b) and (6b), and indicate their size in the table.
Finally, it would be useful if authors: (1) could mention more carefully the limitations of this combined technique; (2) since they have applied to PCT looking for a patent of this idea, it would be nice to give an idea of the cost difference of this new instrument when compared to a Raman and a FTIR independent instruments.
Author Response
Thank you very much for your review.

Round 2
Reviewer 2 Report
I consider the article fit to be published in its actual form.